# High-Modulus Laminated SiC/AZ91 Material with Adjustable Microstructure and Mechanical Properties Based on the Adjustment of the Densities of the Ceramic Layers

**DOI:** 10.3390/ma16186168

**Published:** 2023-09-12

**Authors:** Zeqi Du, Kunkun Deng, Kaibo Nie, Cuiju Wang, Chao Xu, Quanxin Shi

**Affiliations:** 1Shanxi Key Laboratory of Advanced Magnesium-Based Materials, College of Materials Science and Engineering, Taiyuan University of Technology, Taiyuan 030024, China; 2School of Materials Science and Engineering, Harbin Institute of Technology, Harbin 150001, China

**Keywords:** bio-inspired composites, freeze–casting, metal matrix composites (MMC), mechanical properties

## Abstract

To address the issue of inadequate strength and plasticity in magnesium matrix composites, SiC preforms were prepared using the freeze-casting process. The effects of sintering temperature on the microstructure, mechanical properties, and fracture behavior of SiCp/AZ91 magnesium matrix composites were studied by controlling the density of SiC preforms through low-temperature sintering. The results indicate that as the sintering temperature decreases, the reaction products in the SiC layer decrease, resulting in lower SiC preform density and increased content of AZ91 alloy filling in the layer. The increased alloy content in the ceramic layer not only inhibits crack initiation but also hinders crack propagation, thereby endowing the SiCp/AZ91 laminated material with excellent compressive strength and compressive strain. At the sintering temperature of 900 °C, the SiCp/AZ91 laminated material exhibits impressive compressive strength and strain values of 623 MPa and 8.77%, respectively, which demonstrates an excellent combination of strength and toughness.

## 1. Introduction

Magnesium and its alloys, known as the lightest metallic structural materials, with low density and high specific strength and stiffness, hold immense potential for applications in aerospace, automotive, electronics, and various other industries [1,2]. However, the application of magnesium alloys in these fields is limited by inherent drawbacks, such as low absolute strength and modulus of elasticity [3]. In order to give full play to the advantages of magnesium alloys in the industrial field, researchers commonly employ compounding techniques by incorporating reinforcing particles (such as TiC [4], SiCp [5,6], etc.) into magnesium alloys to produce particulate-reinforced magnesium matrix composites (PMMCs). These composites are characterized by high modulus and strength and exceptional wear resistance. The modulus of PMMCs is closely tied to the amount of reinforcement, with a significant increase in modulus as the reinforcement content increases, albeit at the expense of reduced plasticity [7]. Consequently, the conflicting relationship between modulus and plasticity has emerged as a critical issue to be solved in the development of magnesium matrix composites.

To optimize the influence of the reinforcing phase and matrix on the strength and toughness of composites, researchers have achieved a synergistic improvement by designing conformally shaped preforms and employing an infiltration process [8,9]. Among these methods, the freeze-casting technique provides an effective means of achieving preform conformational modulation [10]. This method is environmentally friendly, with a simple preparation process and a low cost, allowing for the processing of not only large-sized bulk structural materials but also precise control over the microstructure. The method is based on the solidification principle, which involves using a water-based ceramic slurry in directional solidification to align ceramic particles into layers. The subsequent ice-crystal sublimation creates pores within the ceramic, and high-temperature sintering solidifies the ceramic, imparting it with a certain strength [10]. For instance, in the case of SiC preforms, high-temperature sintering involves the reaction of SiC to produce SiO_2_, which then reacts with a sintering aid to form a low-melting-point aluminum silicate. This silicate fills the pores within the SiC particles and facilitates their connection [11]. Previous studies on SiC preforms mainly involve high-temperature sintering to enhance the bonding between SiC particles and achieve a high-strength SiC ceramic scaffold [12,13]. Generally, the higher the strength of the ceramic preform, the more significant the subsequent increase in composite strength after infiltration.

It has been observed that crack initiation in composites is primarily attributed to the ceramic layer, which imparts high strength but also increases the brittleness of the composite [14]. Therefore, enhancing the plasticity of SiCp-reinforced magnesium matrix composites can be achieved by retarding crack initiation within the SiC particles. When the matrix is infused into the ceramic skeleton, it is anticipated that the magnesium matrix composite can alleviate stress concentration within the ceramic layer during the loading process, delay the fracture of the ceramic layer, and consequently improve its plasticity. Based on this concept, Hu et al. filled the pores within a sintered loose TiC ceramic skeleton with a Cu melt, resulting in a 200% increase in crack-initiation toughness and a 160% increase in crack-propagation toughness [15]. By adjusting the wettability of B4C ceramic skeleton and Al solution, Wang et al. conducted a die-casting process to cast the molten alloy into a freeze-cast porous ceramic skeleton. As a result, the flexural strength and fracture toughness of the composite increased by 37% and 69%, respectively [16]. This highlights the feasibility of filling the matrix into the ceramic skeleton to enhance the plasticity of magnesium matrix composites. Additionally, increasing the porosity of the ceramic skeleton by reducing the sintering temperature, thus minimizing the formation of reaction products and creating space for the infiltration of magnesium alloy liquid, is another approach to achieve the filling of the ceramic layer with the alloy [17]. Nonetheless, lowering the sintering temperature leads to increased particle looseness and decreased strength of the preform itself. In the subsequent liquid-infiltration process, the collapse of the ceramic layer can easily occur, which prevents the successful preparation of the material. This demonstrates the clear relationship between sintering temperature, looseness within the ceramic layer, and material forming [18]. Therefore, regulating the sintering temperature to obtain a balance between preform strength and looseness is crucial. An excessively dense ceramic layer undoubtedly impedes the filling of molten metal, while an overly loose ceramic layer cannot withstand the pressure generated during the infiltration process, resulting in preform failure. The regulation of sintering temperature to control looseness within the ceramic layer and achieve high-strength, high-toughness magnesium matrix composites represents a current focus in the preparation of freeze-casting-based composites.

Therefore, the primary purpose of this study is to explore the preparation of magnesium matrix composites with high strength and toughness based on the sintering temperature control of the looseness in the ceramic layer. In this work, layered SiC preforms were prepared through freeze–casting and sintered at 800 °C, 900 °C, and 1000 °C. Subsequently, the AZ91 magnesium alloy was hydraulically pressed into the pores and interlayer gaps of the SiC preform layer under pressure to prepare a SiCp/AZ9 magnesium base material. On this basis, the effects of sintering temperature on the internal structure, microstructure, mechanical properties, and fracture behavior of the laminated SiCp/AZ91 material were studied.

## 2. Materials and Methods

### 2.1. Materials

Industrial SiC (D50 = 5 µm, 99.5% purity, Henan White Pigeon Group Co., Ltd., Henan, China) was used as the primary material, and Al_2_O_3_ (Purity: 99.9%, 5 μm, Taipeng Metal Material Co., Ltd., Jiangsu, China) and Y_2_O_3_ (Purity: 99.9%, 5 μm, Hebei Chuancheng Metal Material Co., Ltd., Hebei, China) were used as sintering aids (molar ratio of Al_2_O_3_ and Y_2_O_3_ was 5:3) [12]. Sodium polymethacrylate (PMMA-Na, 40 wt.% in water solution, Shanghai Macklin Biochemical Co., Ltd., Shanghai, China) was used as a dispersant. Gelatin (Shanghai Macklin Biochemical Co., Ltd., Shanghai, China) was used as an organic binder. The n-butanol solution (Shanghai Macklin Biochemical Co., Ltd., Shanghai, China) was used as a defoaming agent. The metal matrix in the composite was a commercial AZ91 magnesium alloy with the chemical composition shown in Table 1.

### 2.2. Sample Preparation

The SiCp/AZ91 laminated material is infiltrated through pressure infiltration of molten AZ91 into a laminated porous SiC preform prepared using the freeze–casting method. The SiC preforms were first prepared with an initial solid-phase content of 30%. A well-dispersed slurry was made by mixing SiC powder, 10 wt% sintering aid (based on the total powders), and 0.4 wt% polymethacrylic acid sodium (PMMA-Na, as a dispersant) in deionized water and ball milling for 16 h. The slurry was heated to 60 °C, 1.5 wt% gelatine was added as a binder, and the slurry was stirred continuously using a stirring paddle until the gelatine was completely melted; then, the slurry was placed in a vacuum chamber for defoaming, and 1–2 drops of n-butanol solution were added. The freezing platform was also set to −30 °C. The freezing platform consisted of a PTFE mold, a copper plate base, and a chiller, where the chiller provided a stable freezing temperature, the alcohol as used as the freezing medium in contact with the copper plate, and a Teflon cylinder (φ60 mm × 60 mm) was placed on the copper plate as the mold. The slurry prepared in this study was poured into a mold, ensuring direct contact between the slurry and the copper plate located at the bottom of the mold, while leaving an exposed surface in contact with the air at the top. After the curing was completed, it was dried at −50 °C for 4 days in a 20 Pa vacuum to sublimate the ice crystals to obtain a raw blank. The freezing-casting process flow diagram is shown in Figure 1. Finally, the billets were sintered at 600 °C for 1 h in a chamber resistance furnace to remove gelatine and PMMA-Na. The temperature was adjusted to 800 °C, 900 °C, and 1000 °C for 2 h and then cooled to room temperature with the furnace, where the heating and cooling rates were 10 °C/min. For the composite-preparation process, the SiC preforms were preheated to 520 °C in a stainless steel mold, and the AZ91 malted magnesium alloy was forced to penetrate completely into the preform by applying pressure on the press for 10 min. For easy description, the SiC preforms sintered at 800 °C, 900 °C and 1000 °C were all filled with AZ91 matrix based on the above fabrication process, denoted as 800 L, 900 L, and 1000 L, respectively.

### 2.3. Characterization

The microstructure of the material was observed via scanning electron microscopy (SEM, MIRA3 XMU, TESCAN, Brno, Czech Republic), with specimens observed in two planes parallel and perpendicular to the freezing direction, and compositional analysis was performed via energy spectrometry (EDS, OXFORD, Britain, UK). The material phase composition of the composites was analyzed using X-ray diffraction (XRD, Smartlab SE, Rigaku, Japan) with a scanning range (2θ) of 20° to 80° and a scanning speed of 2°/min. The XRD patterns were analyzed and processed using Jade 6.5 software and Origin 9.5 software. The modulus of elasticity was tested using a high-temperature modulus of elasticity and damping tester (RFDA-HTVP1750C, IMCE, Genk, Belgium). Compression and three-point bending experiments were carried out in the MTS-E45.105 tester, where the compression specimens were cut along the freezing direction, with a specimen diameter of φ8 mm and a height of 12 mm, and a compression rate of 0.5 mm/min at room temperature. The three-point bending specimens were 35 mm long, with a cross-section of 4 mm × 3 mm and an indenter loading rate of 0.5 mm/min at room temperature, with a span of 20 mm.

## 3. Results

### 3.1. Microstructure

Figure 2 presents an SEM image of the SiCp/AZ91 laminated material in the parallel-to-freezing direction. The SiC ceramic skeleton is represented by the dark grey layer, while the AZ91 matrix is depicted by the black layer. The SiCp layer of the preforms at different sintering temperatures remained intact without defects after infiltration. The overall lamellar structure of the composites also maintains a long-range ordered arrangement, and there are no significant changes in the thickness of the ceramic layers (Figure 2a–c). In addition, the SEM images (Figure 2d–f) depict the SiC particles within the ceramic layer at temperatures of 800 °C, 900 °C, and 1000 °C, respectively. It can be observed that the SiC particles exhibit loose packing at 800 L, primarily interconnected by the matrix. As the sintering temperature increased to 900 L, the bonding among the SiC particles becomes stronger, resulting in a reduced matrix content within the ceramic layer after infiltration. When the sintering temperature was further increased to 1000 L, most of the SiC particles contacted directly, exhibiting enhanced interparticle bonding and a further decrease in the matrix content within the layer. These observations demonstrate that, while the sintering temperature has minimal impact on the thickness of the ceramic layer, it primarily affects the bonding among the SiC particles within the ceramic layer. Specifically, as the sintering temperature increases, the interparticle bonding among the SiC particles becomes more compact, leading to a reduction in the matrix content within the layer.

To further analyze the microstructure of the composite, Figure 3a presents a longitudinal SEM image of the composite at 1000 L. The EDS surface scan results are shown in Figure 3b, demonstrating uniform penetration of the magnesium liquid into the ceramic layers, as well as filling of pores between the layers. EDS analysis was additionally conducted on the SiCp layer (Figure 3c), revealing the presence of Si, Mg, O, Al, and Y elements. The elemental distribution of Figure 3c is further illustrated in Figure 3d, indicating the presence of Mg elements between the particles within the SiCp layer, confirming the successful infiltration of the SiCp layer with magnesium liquid. Furthermore, a diffuse distribution of Al, Y, and O elements near the particles suggests dispersion of the sintering aids surrounding the SiC particles, serving as bonding agents [19].

Figure 4 presents XRD plots of SiCp/AZ91 composites sintered at different temperatures. The prominent diffraction peaks in the composites correspond to the Mullite phase, along with Mg and SiC. Combined with the EDS analysis in Figure 3c,d, the presence of Al, Y, and O elements within the layers confirms the formation of Mullite phase after completing the sintering of the SiC preform [20]. The analysis suggests that during sintering, the SiC particle surface undergoes oxidation, resulting in the formation of SiO_2_. Subsequently, SiO_2_ reacts with the sintering aid to yield Mullite, which is distributed between the SiC particles according to the following reaction equation [21,22]:(1)$$2SiC+3O_{2}\to {2SiO}_{2}+2CO$$
(2)$$SiC+2O_{2}\to SiO_{2}+CO_{2}$$
(3)$${Al}_{2}O_{3}+SiO_{2}+Y_{2}O_{3}\to Mullite$$

After the sintering of SiCp preforms, the Mullite phase generated melts to form a liquid phase, which fills the interstices between ceramic particles, resulting in enhanced bonding among them and increased bond strength [23]. As a result, the strength of the support is improved. With increasing sintering temperature, more Mullite phase is formed, leading to denser ceramic lamellae. Furthermore, XRD patterns of the infiltrated SiCp/AZ91 composites revealed the presence of small amounts of Mg_2_SiO_4_ and Mg_2_Si. It is presumed that the following reaction occurred at the interface between the matrix and particles [24].
(4)$$4Mg+SiO_{2}\to 2MgO+{Mg}_{2}Si$$
(5)$$2MgO+SiO_{2}\to Mg_{2}SiO_{4}$$

Due to the favorable wettability of SiO_2_ and Mg, the Mg component reacts with the residual SiO_2_ during the infiltration process, resulting in the formation of Mg_2_Si and Mg_2_SiO_4_ [25]. This reaction enhances the interfacial bonding strength between the substrate and the ceramic particles [26]. However, as shown in Figure 4a,b, the diffraction peaks are not prominent. This is likely due to the relatively small amount of SiO_2_ generated through surface oxidation of SiC when sintered at 800 °C and 900 °C, leading to lesser amounts of Mg_2_SiO_4_ and Mg_2_Si formation. Upon increasing the temperature to 1000 °C, the diffraction peak of Mg_2_Si becomes detectable, indicating the formation of the Mg_2_Si phase during sintering at 1000 °C.

### 3.2. Mechanical Properties and Fracture Behavior of the Composites

Figure 5a depicts a typical compressive stress–strain curve for the SiCp/AZ91 laminated material when loaded parallel to the freezing direction. Observations reveal that the variation in sintering temperature had a minimal effect on the compressive strength of the composites. The highest compressive strength of 644 ± 20 MPa was achieved at 1000 L, followed by 900 L and 800 L with compressive strengths of 623 ± 22 MPa and 623 ± 25 MPa, respectively. However, the sintering temperature variation considerably affected the compressive strain of the composites. As the sintering temperature decreased, the compressive strain initially increased and then decreased, with noticeable plastic deformation evident in the stress–strain curve. The best plastic deformation was observed at 900 L, with a compressive strain of 8.77% ± 0.21%, representing a 12% and 32% increase compared to 800 L and 1000 L, respectively. Additionally, the sintering temperature influenced the elastic modulus of the composites. Figure 5b illustrates that the elastic modulus increased by approximately 5 GPa for every 100 °C increase in the sintering temperature. At 800 L, the elastic modulus of the composite was 94 GPa, which increased with compact SiC particle bonding within the layer and lower matrix content. The elastic modulus reached 105 GPa at 1000 L. The enhancement of Young’s modulus was quite pronounced, especially when considering that AZ91 alloys have a modulus of only 43 GPa. This enhancement is primarily attributed to the high Young’s modulus of SiC particles (380–470 GPa) and their robust bonding with the matrix [27]. The upper limit of Young ‘s modulus of composites can be calculated using Equation (6) [28]:(6)$$E_{c}=E_{m}\times \left(1-V_{p}\right)+E_{p}\times V_{p}$$
where *E_c_*, *E_m_*, and *E_p_* are the Young’s moduli of the composites, the matrix, and the reinforcing particles, respectively. The upper limit of the predicted Young’s modulus of the laminated SiCp/AZ91 composites is 144–171 GPa. The obvious experimental value is lower than the theoretical value, which is due to the anisotropy of the composite material, which has a significant effect on its elastic modulus. Therefore, according to Halpin–Tsai Equations (7) and (8), the elastic modulus of the longitudinal section of the layered AZ91/SiC composite is further calculated [29]:(7)$$E=E_{AZ91}(1+2sq\varphi )/(1-q\varphi )$$
(8)$$q=(E_{SiC}-E_{AZ91})/(E_{SiC}+2sE_{AZ91})$$
where *E*, *E_SiC_* and *E*_*AZ*91_ are the elastic modulus of composites, SiC ceramics and AZ91 alloy matrix, respectively; *s* is the aspect ratio of reinforced particles; and *φ* is the volume fraction of reinforced particles. The Young’s modulus of the laminated SiCp/AZ91 composites with a volume fraction of 30% is calculated to be 77–94 GPa when the *s* value is 1. The longitudinal elastic modulus of 800 L, 900 L, and 1000 L were actually 94 GPa, 98 GPa, and 105 GPa, respectively, which are slightly higher than the calculated values. This may be related to the fact that the SiC preform is sintered more compactly, thereby improving the elastic modulus of the composite. However, although the elastic modulus of the composites at 800 °C and 900 °C decreased and a certain strength as sacrificed, the plasticity was improved. It can be seen that the mechanical properties and elastic modulus of the composites can be significantly affected by adjusting the bonding of SiC particles in the ceramic layer via sintering temperature.

Figure 6 displays the compressive fracture surfaces at different sintering temperatures, its surface cracks and holes as indicated by the arrow. For 1000 L, clear interface delamination can be observed, with fewer secondary cracks in the SiCp layer, and some SiC particles are fractured, which can be found on the fracture surface in Figure 6c. The compressed fracture at 900 L exhibits minor interface delamination at the layer interface and extensive secondary cracking on the ceramic layer surface. The phenomenon of interface delamination is weakened, and more pronounced secondary cracking occurs on the SiCp layer surface when the sintering temperature decreases to 800 °C. Additionally, significant transgranular fractures are observed in the AZ91 alloy layer. When the composite is loaded along the freezing direction, the ceramic and metal layers experience simultaneous stress, with the metal layer initially deforming upon stress transfer at the layer interface [14,30]. According to the enlarged image in the SiC layer, there are more cracks in the SiC layer of the composite, while the AZ91 layer is relatively smooth. This is because the composite is prone to stress concentration near the SiC particles during the loading process. The larger internal stress leads to the preferential initiation of cracks near the SiC particles in the ceramic layer, which leads to the formation of holes in the ceramic layer. The SiCp layer primarily bears the load, generating multidirectional stress within its structure, while the alloy within the layer helps alleviate stress concentration in the SiC particles. At 1000 L, the SiC particles within the ceramic layer are tightly bonded, and stress concentration can easily occur between the irregular SiC particles in the ceramic layer. The strength difference among the layers results in uncoordinated deformation under load, leading to interface delamination [31]. The relaxation of stress via the matrix is limited, leading to the fracture of the SiCp layer. At 900 L, the densification of the ceramic layer increases, and the filling of magnesium liquid among SiC particles causes the alloy within the ceramic layer to envelop the SiC particles. This inhibits particle fragmentation, retards crack initiation and propagation within the ceramic layer, and significantly enhances plasticity. However, as the sintering temperature further decreases to 800 °C, the particles in the ceramic layer are too loose, and the bonding between the particles is weakened, resulting in a decrease in the strength of the ceramic skeleton itself. Therefore, compared with 900 L, the SiCp layer has a limited constraint effect on the plastic deformation of the AZ91 alloy layer, does not play a good bearing role, and has difficulty inhibiting the initiation and propagation of cracks [32,33]. As a result, the strength and plasticity of the composites at 800 L are reduced.

## 4. Discussion

In this work, laminated porous SiC preforms were prepared using low-temperature sintering, and the properties of the infiltrated composites were found to be closely related to the sintering temperature. At a sintering temperature of 1000 °C, the composite exhibited low plasticity but a high modulus of elasticity. As the sintering temperature decreased, a better balance between strength and plasticity was achieved, although the modulus of elasticity decreased slightly. It can be observed in Section 3.1 that the sintering temperature had no effect on the configuration of the laminate but primarily influenced the bonding among SiC particles within the SiCp layer. Higher sintering temperatures resulted in more compact bonding between SiC particles and reduced matrix content within the layer. When the SiCp/AZ91 laminated material is subjected to load, the AZ91 alloy layer helps relieve stress concentration at the interface, while the ceramic layer bears the load and restrains the plastic deformation of the alloy layer, leading to a synergistic increase in strength and plasticity. However, due to the large difference in strength between the layers, uncoordinated deformation at the interface can result in stress concentration. By increasing the porosity within the ceramic layer through low-temperature sintering, more AZ91 alloy matrix material can be filled among the SiC particles, thereby reducing the strength disparity between the ceramic layer and the AZ91 alloy layer.

### 4.1. Flexural Strength and Toughness

In order to further investigate the bonding strength among particles within the layers and the influence of the AZ91 layers on their fracture properties, bending experiments were conducted. Figure 7a presents a typical three-point flexural stress–strain curve for the SiCp/AZ91 laminated material. It can be observed that as the sintering temperature increases, the flexural strength and flexural strain of the composite decrease. At 800 L, the composite exhibits a flexural strength of 533 ± 27 MPa and a flexural strain of 1.08 ± 0.09%. These values represent a 31% increase in flexural strength and a 47% increase in flexural strain as compared to the 1000 L, as shown in Figure 7b. During bending loading, the difference in deformation capacity becomes more pronounced at higher sintering temperatures due to the distinct deformation capacities of the ceramic and metal layers [34]. The 800 L exhibits higher flexural strengths and flexural strain, which are attributed to the microstructure of the SiCp layers. Generally, the nucleation of the crack means that the material begins to break. The sprouting of cracks happens when the stress field at the crack tip surpasses a critical value [35]. As the sintering temperature decreases, the particles within the ceramic layer become looser, allowing for the infiltration of more AZ91 alloy into the ceramic layer. The alloy layer and the matrix within the ceramic layer hinder crack initiation and propagation, thereby reducing the crack nucleation rate [16]. Accordingly, the flexural strength and flexural strain of the SiCp/AZ91 laminated material increase as the sintering temperature decreases.

### 4.2. Influence of Sintering Temperature on the Flexural Fracture Behavior of the Composites Influence 

To gain further insight into the impact of the ceramic and AZ91 alloy layers on crack initiation and propagation, the flexural fracture and crack-extension paths of the composites were examined. Figure 8 illustrates the lateral fractures of the bent specimens at different sintering temperatures: 800 °C, 900 °C, and 1000 °C (Figure 8a,e,i). At a sintering temperature of 800 °C, the crack-extension direction exhibits a significant deviation from the loading direction, with a crack deflection width of 500 μm. This indicates a higher consumption of fracture energy during the fracture process, resulting in a substantial reduction in the crack-extension rate [36]. As the sintering temperature increases to 900 °C, the crack deflection width decreases to 320 μm, indicating lower energy absorption during crack propagation. At a sintering temperature of 1000 °C, the crack extension appears flatter, with a narrower deflection width of only 230 μm. It is evident that the sintering temperature significantly influences the behavior of the crack extension.

The influence of the sintering temperature on crack extension is directly associated with the microstructure within the ceramic layer [37]. Consequently, this work focuses on analyzing the intra-layer cracking behavior. Observations of Figure 8b,c reveal that cracks primarily concentrate near the ceramic particles at 800 L. Due to lower stress concentration within the layer, the path of crack expansion is obstructed by SiC particles, leading to SiC particle fragmentation and improved fracture resistance of the composite. Furthermore, at 800 L, the composite exhibits a well-bonded interface with no evident interfacial cracks (Figure 8d). These findings indicate that the ceramic layer and the metal layer undergo coordinated deformation during loading [38]. As the strain increases to a certain extent, the ceramic layer is torn, resulting in the formation of numerous parallel branch cracks in addition to the main crack. Additionally, the tear ridge and dimple are observed, illustrating that the AZ91 alloy within the ceramic layer acts as a toughening agent during the fracture process [15]. Therefore, the 800 L composite demonstrates excellent flexural strength and fracture toughness. Furthermore, Figure 8f,j shows the presence of numerous microcracks within the ceramic layer at 900 L, along with some broken SiC particles. Simultaneously, the interface debonding phenomenon at the interface between the particles and the matrix in the layer is weakened, while the branching cracks within the layer are also minimized (Figure 8h). It is evident that intra-layer cracking exhibits distinct patterns depending on the sintering temperature, with microcracking more likely to occur at the particle–matrix interface within the layer at lower sintering temperatures. This is attributed to low-temperature sintering, making SiC particles looser between each other, which is beneficial for filling more alloy. This increased alloy content not only relieves stress concentration but also leads to the sprouting of microcracks at multiple locations within the SiC layer [39]. As the temperature increases, SiC particles in the layer are more dense, which leads to the increase in stress concentration in the layer, and the cracks more easily connect with each other to form the main crack. Simultaneously, as the difference in strength between the layers increases, stress concentrations tend to occur at the interface, leading to interface delamination. When the temperature is raised to 1000 °C, SiC particles are compactly bonded to each other. Upon crack sprouting into the ceramic layer, the ceramic layer undergoes brittle fracturing, and transgranular fractures of the SiC ceramic particles are visible on the fracture surface (Figure 8j). Additionally, due to increased stress concentration among the particles, cracks propagate as continuous main cracks within the composite.

In summary, the 800 L composite material has a certain inhibitory effect on crack expansion due to the multiple sprouting of cracks in the process of deformation to relieve stress concentration, thus consuming more energy for crack expansion. As a result, more pronounced crack deflection occurs, and better plasticity is achieved. With the increase in temperature, the SiC particles within the layer are tightly bound. The inhibition of crack extension by the matrix within the layer is weakened, which favors the formation of the main crack and leads to a weakening of the crack-deflection phenomenon. When the sintering temperature reaches 1000 °C, the number of matrices within the layer decreases and the crack deflection phenomenon is weakened, resulting in a straighter crack extension path [40,41]. The crack pattern undergoes a transition that is mainly influenced by the ratio of restrained metal yield stress (σ_m_) to ceramic strength (σ_c_) and the ratio of the thickness of the alloy layer (t_m_) to the thickness of the ceramic layer (t_c_) [42]. Specifically, multiple-cracking extension occurs when t_m_/t_c_ exceeds the critical value (t_m_/t_c_)_crit_, and the critical value (t_m_/t_c_)_crit_ increases with the increase in the alloy layer/ceramic layer elastic modulus ratio (E_m_/E_c_) [43,44]. As mentioned above, as the sintering temperature increases, only a small amount of alloy liquid fills the ceramic layer into the denser lamellar structure, and a large number of ceramic particles are interconnected with each other, gradually increasing the strength (σ_c_) of the ceramic layer and also gradually increasing the modulus of elasticity (E_c_), thus making the critical value (t_m_/t_c_)_crit_ gradually increase. In this study, as the sintering temperature had no significant effect on the thickness of the alloy layer (t_m_) versus the thickness of the ceramic layer (t_c_), the stress at the crack tip could not be effectively dispersed by the AZ91 alloy layer due to the high critical value (t_m_/t_c_)_crit_ of the sample at 1000 L and the large strength difference between the ceramic and alloy layers, which together led to the crack-extension pattern exhibiting a single crack-extension mode [34]. Therefore, lowering the sintering temperature can promote a shift from a single-crack pattern to a multiple-crack pattern, thereby increasing the fracture toughness.

The presence of alloy layers in the composite can reduce stress concentrations within the ceramic layer, leading to delayed fracture and improved toughness of the composite plate [45]. As cracks in various regions continue to propagate and penetrate the ceramic layer, the interface acts as a barrier to further crack propagation [46]. Under increasing stress, the alloy and ceramic layers undergo plastic deformation together. This not only relieves stress concentration within the composite but also reduces stress at the interface. As the interface hinders the expansion of the main crack, new microcracks form within the ceramic layer, thus dispersing and absorbing energy. This increases the driving force for the expansion of the main crack, ultimately enhancing the fracture toughness of the material [47]. As demonstrated in Figure 8c,j,k, multiple cracks are initiated at 800 L, resulting in limited crack propagation energy and relatively strong inhibition of crack extension by the alloy layer. As the temperature increases, the development of main cracks within the layer becomes more prominent, leading to higher energy at the crack tip and weaker inhibition of crack propagation by the alloy layer. Based on the aforementioned analysis, it is evident that the infiltration of a significant amount of magnesium alloy within the low-temperature sintered alloy layer is beneficial for enhancing its plasticity.

Based on the aforementioned findings, the diagram of the fracture mechanism is presented in Figure 9. In the laminated composite, the critical stress required for crack initiation is lower in the ceramic layer, which is more brittle in comparison to the AZ91 alloy layer. Consequently, cracks tend to develop preferentially within the ceramic layer and subsequently propagate along the loading direction within the layer. Particle-dense areas are susceptible to stress concentrations, resulting in crack initiation and subsequent expansion within the ceramic layer in the direction of loading and fracture [48]. Decreasing the sintering temperature promotes the propagation of microcracks within the ceramic layer and alters the single-crack propagation pattern. When the cracks contact the AZ91 alloy layer, the interface begins to deform and absorb fracture energy, thus impeding further crack propagation. With the increase in temperature, the density of the SiC layer increases, which makes it easier to form main cracks in the layer. Under a load, the composite material easily breaks due to the fracture of the ceramic layer, which leads to the crack propagating in the ceramic layer of the material and passing through the AZ91 alloy layer, eventually leading to material failure.

## 5. Conclusions

In this work, the SiC preforms with a volume fraction of 30% were prepared using the freeze–casting method, and the influences of sintering temperature on the microstructure, mechanical properties, and fracture behavior of the SiCp/AZ91 laminated material were investigated based on the regulation of the density of the SiC layer by the sintering temperature, and the following main conclusions were drawn:(1)By lowering the sintering temperature, the formation of reaction products between the SiC preform and the SiC/AZ91 composite is reduced, effectively reducing the looseness of SiC particles within the ceramic layer. This enables the filling of the Mg matrix within the ceramic skeleton.(2)Reducing the density of SiC preforms does not significantly enhance the compressive strength of the composite, but it has a significant impact on the compressive strain. In the SiCp/AZ91 laminated material, the ceramic skeleton exhibits certain strength, while the hard SiC particles within the ceramic layer are encapsulated by the soft matrix at 900 L. This combination not only provides the composite with high strength but also retards crack initiation within the ceramic layer. The compressive strength and compressive strain of the composite reach 623 ± 22 MPa and 8.77% ± 0.21%, respectively, representing the best strength–toughness combination.(3)Reducing the density of SiC preforms is beneficial to improve the flexural strength and flexural strain of the composites. In the SiCp/AZ91 laminated material, the flexural strength and flexural strain increased by 31% and 47%, respectively, at 800 L compared to 1000 L.(4)By reducing the sintering temperature, the initiation of cracks within the SiC particles can be retarded. The matrix filling within the ceramic layer brings about various toughening mechanisms, including synergistic plastic deformation of the ceramic/alloy layers, crack deflection, and multiple crack extension modes.

## Figures and Tables

**Figure 1 materials-16-06168-f001:**
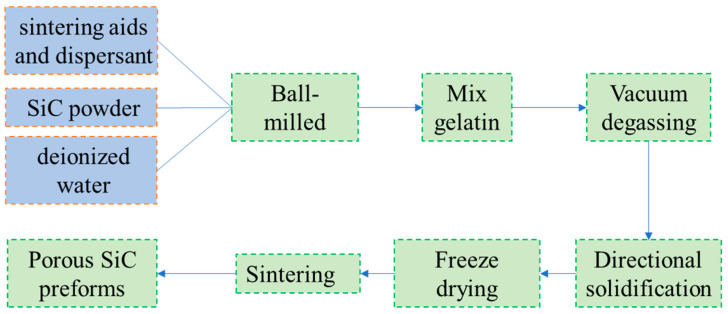
Flow chart of freeze–casting process.

**Figure 2 materials-16-06168-f002:**
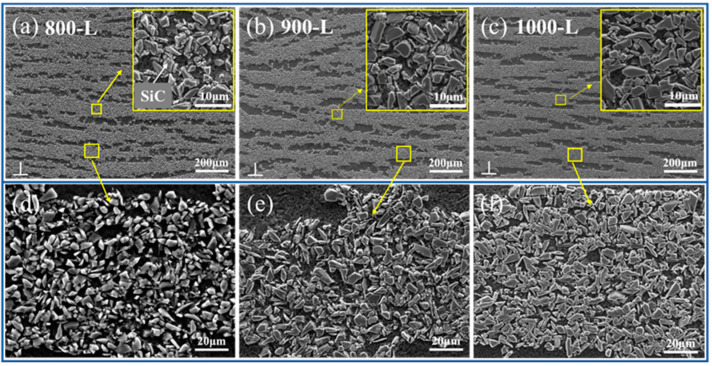
(**a**–**c**) SEM images of the cross-section of the composite at 800 L, 900 L, and 1000 L (**d**–**f**), corresponding to the magnified SEM images of the particles within the ceramic layer in (**a**–**c**).

**Figure 3 materials-16-06168-f003:**
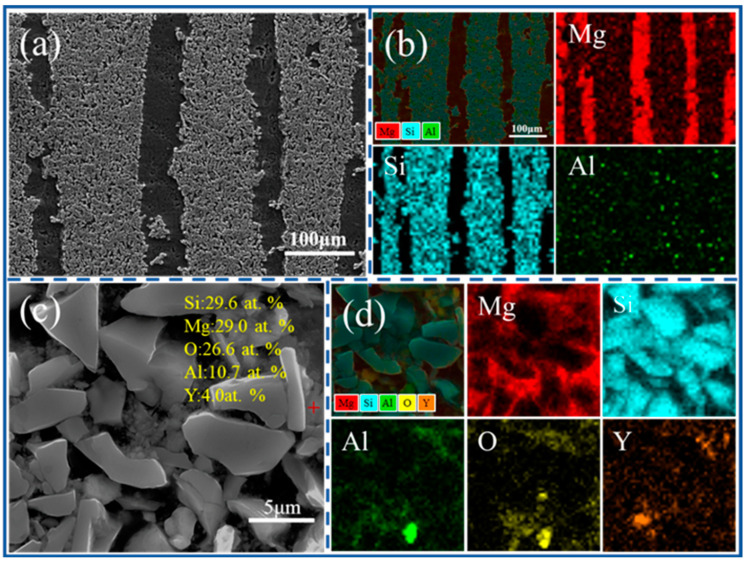
(**a**,**b**) SEM images of the longitudinal section of the composite at 1000 L and its surface scan. (**c**) Typical phase EDS analysis of the composite at 800 L. (**d**) Surface scan of the composite at 800 L.

**Figure 4 materials-16-06168-f004:**
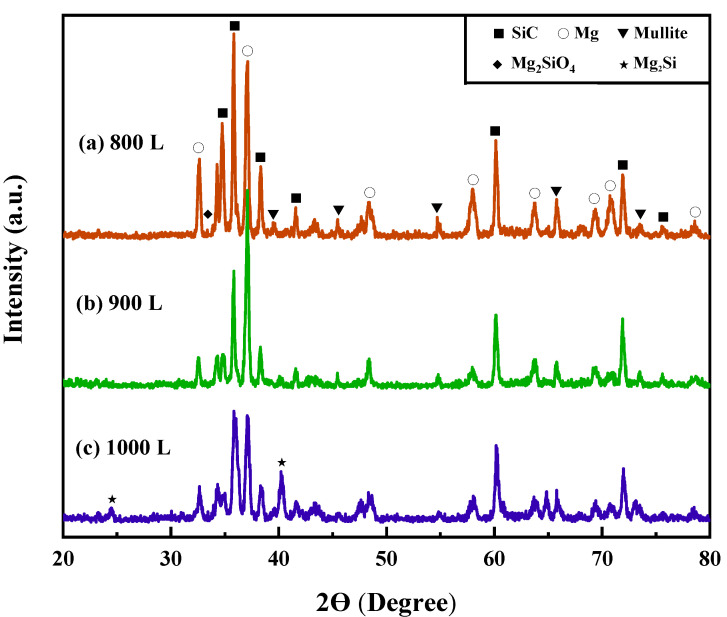
XRD images at different sintering temperatures: (**a**) 800 L, (**b**) 900 L, (**c**) 1000 L.

**Figure 5 materials-16-06168-f005:**
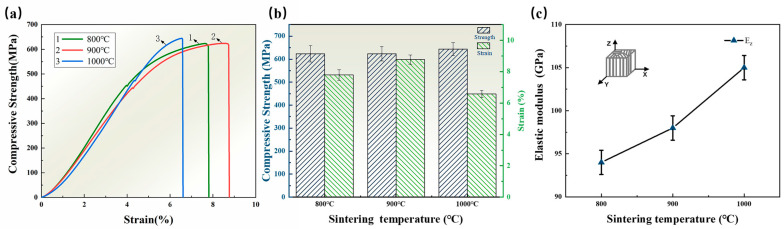
Longitudinal section of the SiCp/AZ91 laminated material at different sintering temperatures: (**a**) Compressive stress–strain curve; (**b**) compressive stress–strain bar graph; (**c**) modulus of elasticity.

**Figure 6 materials-16-06168-f006:**
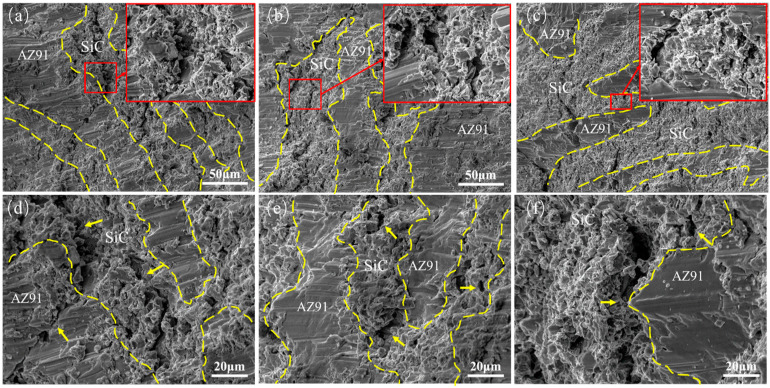
Typical fracture surfaces after compression at room temperature: (**a**,**d**) 800 °C, (**b**,**e**) 900 °C, (**c**,**f**) 1000 °C.

**Figure 7 materials-16-06168-f007:**
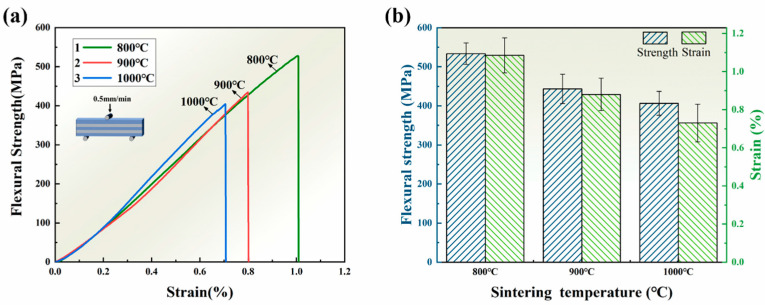
Flexural stress–strain curves: (**a**) flexural stress–strain curves, (**b**) histograms for the SiCp/AZ91 laminated material with different sintering temperatures.

**Figure 8 materials-16-06168-f008:**
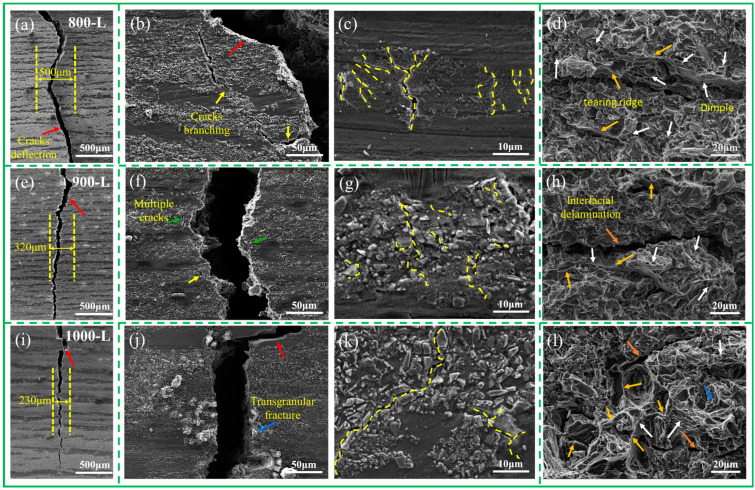
(**a**–**d**) show the crack-expansion path, local microcrack, and fracture shape of the composite at 800 L; (**e**–**h**) show the crack-expansion path, local microcracks, and fracture profiles of the composite at 900 L, (**i**–**l**) are the crack expansion path, local microcrack, and fracture profile of the composite at 1000 L.

**Figure 9 materials-16-06168-f009:**
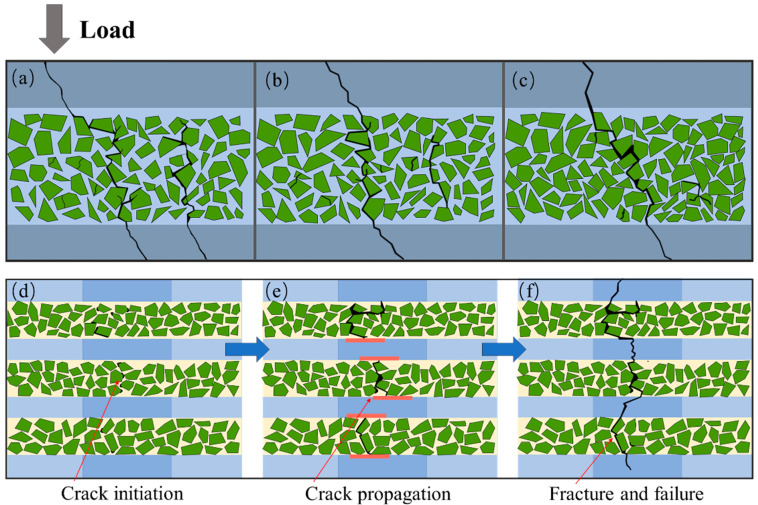
(**a**–**c**) show the fracture mechanism within the layers of the composites at 800 L, 900 L and 1000 L; (**d**–**f**) show the crack-sprouting and the extension of the interlayer AZ91/SiC composites.

**Table 1 materials-16-06168-t001:** AZ91 alloy main chemical components.

Material	Al	Zn	Mn	Fe	Si	Cu	Ni	Mg
AZ91	8.500	0.800	0.300	0.004	0.050	0.025	0.010	Bal.

## Data Availability

Data will be made available on request.

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
